# *TOMM40* Genetic Variants Cause Neuroinflammation in Alzheimer’s Disease

**DOI:** 10.3390/ijms24044085

**Published:** 2023-02-17

**Authors:** Yi-Chun Chen, Shih-Cheng Chang, Yun-Shien Lee, Wei-Min Ho, Yu-Hua Huang, Yah-Yuan Wu, Yi-Chuan Chu, Kuan-Hsuan Wu, Li-Shan Wei, Hung-Li Wang, Ching-Chi Chiu

**Affiliations:** 1Department of Neurology, Chang Gung Memorial Hospital Linkou Medical Center, College of Medicine, Chang Gung University, Taoyuan 33302, Taiwan; 2Dementia Center, Taoyuan Chang Gung Memorial Hospital, Taoyuan 33378, Taiwan; 3Department of Laboratory Medicine, Chang Gung Memorial Hospital Linkou Medical Center, Taoyuan 33305, Taiwan; 4Department of Medical Biotechnology and Laboratory Science, Chang Gung University, Taoyuan 33302, Taiwan; 5Department of Biotechnology, Ming Chuan University, Taoyuan 33348, Taiwan; 6Genomic Medicine Research Core Laboratory, Chang Gung Memorial Hospital, Taoyuan 33305, Taiwan; 7Neuroscience Research Center, Chang Gung Memorial Hospital at Linkou, Taoyuan 33305, Taiwan; 8Healthy Aging Research Center, College of Medicine, Chang Gung University, Taoyuan 33302, Taiwan; 9Department of Physiology and Pharmacology, College of Medicine, Chang Gung University, Taoyuan 33302, Taiwan; 10Division of Movement Disorders, Department of Neurology, Chang Gung Memorial Hospital at Linkou, Taoyuan 33305, Taiwan

**Keywords:** Alzheimer’s disease, TOMM40, SNP, microglia, neuroinflammation, NLRP3, NF-κB, hippocampal neurons

## Abstract

Translocase of outer mitochondrial membrane 40 (*TOMM40*) is located in the outer membrane of mitochondria. *TOMM40* is essential for protein import into mitochondria. *TOMM40* genetic variants are believed to increase the risk of Alzheimer’s disease (AD) in different populations. In this study, three exonic variants (rs772262361, rs157581, and rs11556505) and three intronic variants (rs157582, rs184017, and rs2075650) of the *TOMM40* gene were identified from Taiwanese AD patients using next-generation sequencing. Associations between the three *TOMM40* exonic variants and AD susceptibility were further evaluated in another AD cohort. Our results showed that rs157581 (c.339T > C, p.Phe113Leu, F113L) and rs11556505 (c.393C > T, p.Phe131Leu, F131L) were associated with an increased risk of AD. We further utilized cell models to examine the role of TOMM40 variation in mitochondrial dysfunction that causes microglial activation and neuroinflammation. When expressed in BV2 microglial cells, the AD-associated mutant (F113L) or (F131L) TOMM40 induced mitochondrial dysfunction and oxidative stress-induced activation of microglia and NLRP3 inflammasome. Pro-inflammatory TNF-α, IL-1β, and IL-6 released by mutant (F113L) or (F131L) TOMM40-activated BV2 microglial cells caused cell death of hippocampal neurons. Taiwanese AD patients carrying TOMM40 missense (F113L) or (F131L) variants displayed an increased plasma level of inflammatory cytokines IL-6, IL-18, IL-33, and COX-2. Our results provide evidence that TOMM40 exonic variants, including rs157581 (F113L) and rs11556505 (F131L), increase the AD risk of the Taiwanese population. Further studies suggest that AD-associated mutant (F113L) or (F131L) TOMM40 cause the neurotoxicity of hippocampal neurons by inducing the activation of microglia and NLRP3 inflammasome and the release of pro-inflammatory cytokines.

## 1. Introduction

Alzheimer’s disease (AD), characterized by selective neurodegeneration in brain regions involved in emotional and cognitive function, is the most prevalent cause of dementia among older people [1]. Genetic variants increase the risk of developing AD [2,3,4]. The *Apolipoprotein E* (*APOE*) gene located at chromosome 19q13.32 is the strongest risk factor for AD and accounts for approximately 50% of the total risk contribution [5,6,7,8,9,10]. However, patients carrying *APOE* variants do not necessarily develop AD. Genome-wide association studies revealed that genetic variants neighboring APOE loci also increase AD risk [11,12]. Surrounding genes of the *APOE* loci, such as *TOMM40* (Translocase of outer mitochondrial membrane 40), *PVRL2*, and *APOC1*, display a strong linkage disequilibrium in the *APOE* region and could also be involved in the pathogenesis of AD [13,14,15].

The *TOMM40* gene is located adjacent to the 5′-upstream of the APOE gene and is one of the *APOE*-surrounding genes. TOMM40 is a channel-forming subunit of the mitochondrial TOMM complex that is required for protein import into mitochondria [16]. Many studies suggest that the *TOMM40* gene may contribute to AD risk [17,18]. Single-nucleotide polymorphisms (SNPs) within the *TOMM40* gene are associated with amyloid deposition and influence the metabolism of amyloid beta peptide (Aβ) [19,20,21]. Poly-T repeats’ polymorphism within intron 6 (rs10524523) of the *TOMM40* gene has been shown to affect AD onset age and contribute to AD susceptibility by regulating the expression of *APOE* and *TOMM40* genes [22,23,24]. Differential transcription of *TOMM40* RNA in the brain has been shown to be an indicator of mitochondrial dysfunction in AD [17]. The majority of *TOMM40* genetic variants that were associated with AD susceptibility often reside in noncoding regions with unclear functions [25]. In AD, the related quantitative traits of functional genetic variants of *TOMM40* are still unclear. The pathogenic mechanism by which *TOMM40* genetic variants increase the risk of AD remains unknown.

Microglial-mediated neuroinflammation, the inflammatory response of CNS, is involved in the etiopathogenesis of AD [26,27]. Under physiological status, microglia, which act as resident macrophages of CNS, mediate the development of CNS and regulate immune responses in CNS [28]. In the presence of pathological or inflammatory stimuli, microglia change from a resting state to an activated state and secrete pro-inflammatory cytokines, including interleukin-1β (IL-1β), interleukin-6 (IL-6), and tumor necrosis factor-α (TNF-α), which subsequently cause the degeneration of CNS neurons [26,27,29]. Under pathological or inflammatory conditions, a crucial step for damage-associated molecular pattern (DAMP)- or pathogen-associated molecular pattern (PAMP)-induced microglial activation is the oligomerization and activation of the microglial NLRP3 inflammasome complex, which is composed of nucleotide-binding oligomerization domain and leucine-rich-repeat-and pyrin-domain-containing 3 (NLRP3), the apoptosis-associated speck-like protein containing a CARD (ASC), and pro-caspase-1 [30,31,32,33]. The dysregulated overactivity of the microglial NLRP3 inflammasome resulting from microglial activation is involved in the pathogenesis of AD [34,35,36].

Mitochondria are a major source of intracellular ROS and are sensitive to oxidative stress [37]. Mitochondrial malfunction is involved in the pathogenesis of AD [38]. TOMM40 is vital for maintaining mitochondrial function and is involved in the influx of proteins and Aβ into mitochondria [16,39,40]. In the brains of AD patients, Aβ is accumulated in the mitochondrial import channel and causes mitochondrial dysfunction [41]. Impaired TOMM40-mediated protein transport of mitochondria could lead to the accumulation of Aβ in mitochondrial cristae, which results in mitochondrial malfunction and the overproduction of ROS [42]. Increased formation of mitochondrial ROS activates the mitogen-activated protein kinase (MAPK) cascade, resulting in microglial activation [43]. Moreover, the overproduction of ROS activates the nuclear factor-κB (NF-κB) pathway, which causes activation of the NLRP3 inflammasome and neuroinflammation [44]. Therefore, it is possible that mutations of TOMM40 could cause mitochondrial malfunction and oxidative stress, resulting in microglial activation and increased risk of AD.

In this study, we identified genetic variants of the *TOMM40-APOE* region and determined the significance of *TOMM40* variants in Taiwanese AD patients. Missense variants within the *TOMM40* gene, rs157581 (c.339T > C, p.Phe113Leu) and rs11556505 (c.393C > T, p.Phe131Leu), were associated with increased risk of AD. Our results also demonstrated that mutant (F113L) or (F131L) TOMM40 caused activation of the NLRP3 inflammasome and microglia, leading to the release of pro-inflammatory cytokines and resulting in the cell death of hippocampal neurons.

## 2. Results

### 2.1. AD Patients Exhibit Genetic Variants within TOMM40 Gene

A high-density whole-genome association study showed strong associations between SNPs neighboring the *APOE* loci and AD risk [11]. Although *APOE* is a strong risk factor for AD, it is believed that additional factors with the *APOE* locus contribute to the pathogenesis of AD [18,45,46]. The *TOMM40* gene is within *APOE*-surrounding regions and is located in proximity to the *APOE* gene [18,46]. DNA samples from 80 Taiwanese AD patients were examined using a targeted panel of whole-genome sequencing. When compared with the 1000 Genome Databases, genetic hotspots within the *TOMM40-APOE* region were identified as AD susceptibility (Table 1). Exonic variants included rs772262361 (a synonymous variant within *TOMM40*, c.198A > G, p.Ser66=), rs157581 (a missense variant in *TOMM40*, c.339T > C, p.Phe113Leu), rs11556505 (a missense variant within *TOMM40*, c.393C > T, p.Phe131Leu), and rs440446 (a missense variant in *APOE*, p.Asn14Lys) (Table 1). SNP rs772262361, located in the CpG-rich loci, was a novel mutation (Appendix A). Intronic variants, including rs184017, rs2075650, and rs157582, were found in Taiwanese AD patients (Table 1). We further replicated the association of the three *TOMM40* functional hotspots, including rs772262361, rs157581, and rs11556505, with AD susceptibility using another set of AD patients and controls.

### 2.2. Exonic SNP of TOMM40 Are Linked to Increased AD Susceptibility

The frequency of rs772262361 in Taiwan was further examined in normal controls (NC) of Taiwan Biobank (controls). All three functional variants (rs772262361, rs157581, and rs11556505) were evaluated in 213 normal controls (NC) that were ascertained in this study, 393 AD patients, and 1025 controls from Taiwan Biobank. The genotype and allele frequency of *TOMM40* among NC, AD patients, and controls were displayed in Table 2. SNPs were considered in Hardy–Weinberg equilibrium at a significance level of 0.05. Two *TOMM40* SNPs, rs157581 and rs11556505, were significantly associated with AD.

### 2.3. AD-Associated TOMM40 Genetic Variants, but Not Wild-Type TOMM40, Causes Mitochondrial Dysfunction and Oxidative Stress of Microglial Cells

In this study, we hypothesized that the AD-associated genetic mutation of TOMM40 causes mitochondrial malfunction and oxidative stress in microglia. To test this hypothesis, WT TOMM40 and AD-associated TOMM40 genetic variants, (F113L) and (F131L) TOMM40, were transiently expressed in BV2 microglial cells (Figure 1A). Compared to control cells or cells expressing WT TOMM40, a reduction in the fluorescence intensity of TMRM, which is a dye of mitochondrial membrane potential (ΔΨm), and ΔΨm depolarization was found in BV2 microglial cells transfected with cDNA of (F113L) or (F131L) TOMM40 (Figure 1B). Imaging analysis of MitoSox showed that the fluorescence level of MitoSox and the mitochondrial level of superoxide, which is a major ROS, were significantly upregulated in BV2 microglial cells expressing (F113L) or (F131L) TOMM40 (Figure 1C).

### 2.4. AD-Associated TOMM40 Genetic Variants Cause Microglial Activation

Mutant (F113L) or (F131L) TOMM40-induced mitochondrial malfunction and oxidative stress (Figure 1B,C) could lead to the activation of microglial cells. In accordance with this hypothesis, immunofluorescence imaging staining demonstrated that compared with control cells or cells expressing WT TOMM40, upregulated protein expression of microglial protein marker ionized calcium-binding adaptor molecule 1 (Iba-1) was observed in BV2 microglia cells transfected with cDNA of (F113L) or (F131L) TOMM40 (Figure 1D).

### 2.5. Mutant (F113L) or (F131L) TOMM40 Activates NF-κB Cascade and NLRP3 Inflammasome in Microglial Cells

In this study, it was hypothesized that (F113L) or (F131L) TOMM40-induced malfunction and oxidative stress of mitochondria leads to the activation of the NF-κB signaling cascade and the NLPR3 inflammasome in microglial cells. Consistent with this hypothesis, immunoblotting assays showed that the expression of (F113L) or (F131L) TOMM40 in microglial cells increased the protein expression of phospho-IKKα/β^Ser176/180^ and active phospho-NF-κB p65 in BV2 microglial cells (Figure 2A). Immunoblotting assays demonstrated that mutant (F113L) or (F131L) TOMM40 caused the activation of the NLRP3 inflammasome by upregulating protein levels of NLRP3 and ASC and cleaved active caspase-1 in BV2 microglial cells (Figure 2B).

### 2.6. AD-Associated TOMM40 Genetic Variants Cause the Release of Pro-Inflammatory Cytokines from Microglia Cells, Leading to Cell Death of Hippocampal Neurons

NLPR3 inflammasome activation in microglial cells results in the overproduction and release of pro-inflammatory cytokines, leading to neuronal loss [40]. ELISA analysis indicated that the levels of pro-inflammatory cytokines, including IL-1β, IL-6, and TNF-α, were significantly increased in the culture medium (CM) from BV2 microglial cells expressing (F113L) or (F131L) TOMM40 (Figure 3A). To provide evidence that pro-inflammatory cytokines released from BV2 cells expressing (F113L) or (F131L) TOMM40 cause neuronal death, the CM of HT22 hippocampal neuronal cells was replaced with the CM of BV microglia transfected with cDNA of WT, (F113L), or (F131L) TOMM40. As shown in Figure 3B, the CM of BV2 microglial cells expressing (F113L) or (F131L) TOMM40 significantly decreased the cell viability of HT22 hippocampal neuronal cells.

### 2.7. Plasma Levels of IL-6, IL-18, IL-33, and COX-2 Are Upregulated in AD Patients Carrying TOMM40 Genetic Variants

According to our hypothesis that AD-associated TOMM40 genetic variants cause microglial activation, plasma levels of cytokines and COX-2 are expected to be upregulated in AD patients carrying TOMM40 genetic variants. ELISA assays demonstrated that plasma levels of IL-6, IL-18, IL-33, and COX-2 were significantly increased in AD patients carrying TOMM40 genetic variants, including rs772262361, rs157581, and rs11556505 (Figure 4 and Table 3).

## 3. Discussion

The *APOE* gene located at chromosome 19q13.32 is a strong risk gene for AD and accounts for approximately 50% of AD cases [5,6]. APOE is involved in the clearance and aggregation of the amyloid-β peptide and tau neurofibrillary degeneration [8]. For AD pathology, APOE has been shown to be co-localized with cholesterol and fibrillary Aβ in neuritic plaques and neurofibrillary tangles [47]. *APOE* genotypes are linked to lipid homeostasis and neuroinflammation [48]. *APOE ε2/ε3/ε4* alleles are haplotypes constructed by two missense variants, rs7412 and rs429358 [49]. The *APOE ε4* allele is the strongest risk factor for AD in African American and Caucasian populations [50]. Compared with AD patients in western countries, the frequency of the *APOE ε4* allele is lower in the Asian population [51]. The incidence of AD is similar between Caucasian and Chinese populations [52]. Other genetic modifiers are likely to contribute to the pathogenesis of AD in Asian patients [46,53]. Genetic variants in the *TOMM40*-*APOE* locus could increase the susceptibility of AD [18,46,54,55,56]. The *TOMM40* gene adjacent to the *APOE* gene is in strong linkage disequilibrium with *APOE* [57]. Mutations within the *TOMM40* gene are implicated in the increased risk and pathogenesis of AD [58,59,60,61]. Several functional SNPs within the *TOMM40* gene are identified in late-onset AD (LOAD) [22]. In the Asian AD population, *APOE* and *TOMM40* variants synergistically increase the risk of AD [46]. Therefore, *TOMM40* could be an important risk factor for AD in the Taiwanese AD population.

In this study, we hypothesized that mutations of the *TOMM40* gene contribute to increased AD susceptibility in the Taiwanese population. To test this hypothesis, NGS analysis was performed to identify the genetic variants with the *TOMM40* locus of Taiwanese AD patients. In this study, exonic variants, including rs772262361, rs157581 (c.339T > C, p.Phe113Leu), rs11556505 (c.393C > T, p.Phe131Leu), and intronic variants, including rs184017, rs2075650, and rs157582, were identified in Taiwanese AD patients. Further genetic association studies suggest that two exonic SNPs of TOMM40, rs157581 and rs11556505, are linked to increased AD risk in the Taiwanese population. Previous studies reported that *TOMM40* poly-T repeats polymorphism within intron 6 (rs10524523) decreases the onset age of AD and contributes to increased AD susceptibility in Caucasian populations by regulating the expression of *TOMM40* and *APOE* transcription [23,62]. *TOMM40* SNPs, including rs157580, rs2075650, and rs157581, increase the AD risk in Canadian and Italian populations [59,61]. *TOMM40* rs10524523 is associated with decreased volume of gray matter and impaired cognition in AD patients [63]. *TOMM40* SNPs, rs10524523 and rs2075650, are statistically related to cognitive function, brain integrity, and the alternation of the inflammatory pathway [64,65].

Microglia, which are innate immune cells of CNS, play a vital role in clearing pathogenic molecules and mediating the neuroinflammatory reaction [26,66]. Under pathological conditions, PAMP or DAMP induces the activation of microglia and NLRP3 inflammasome and resulting in neuroinflammation [34,67]. Activated microglia release pro-inflammatory cytokines, including TNF-α, IL-1β, and IL-6, and cause neuronal death [68,69]. Microglial activation-induced neuroinflammation is one of the important mechanisms underlying the pathogenesis of AD [26,70,71]. Several lines of evidence suggest that mitochondrial malfunction of microglia induces microglial activation and is involved in AD pathogenesis [66]. The accumulation of Aβ within mitochondria causes mitochondrial dysfunction and oxidative stress [70,72]. TOMM40 is essential for normal mitochondrial function by mediating the import of proteins, including Aβ, into mitochondria [40,73]. In this study, it was hypothesized that impaired function of AD-associated mutant TOMM40 including (F113L) and (F131L) TOMM40 could cause the accumulation of mitochondrial Aβ and result in mitochondrial malfunction and oxidative stress of microglia, leading to the activation of microglia and the NLRP3 inflammasome, the release of neurotoxic pro-inflammatory cytokines, and the subsequent cell death of hippocampal neurons. In accordance with our hypothesis, the expression of AD-associated TOMM40 genetic variants, (F113L) and (F131L) TOMM40, caused mitochondrial dysfunction by reducing mitochondrial membrane potential and oxidative stress by increasing the mitochondrial level of superoxide in BV2 microglial cells. AD-associated (F113L) or (F131L) TOMM40-induced mitochondrial malfunction and oxidative stress lead to the activation of the NLRP3 inflammasome and inflammatory NF-κB pathway in BV2 microglial cells. Furthermore, pro-inflammatory TNF-α, IL-1β, and IL-6 released by mutant (F113L) or (F131L) TOMM40-activated BV2 microglial cells cause the cell death of hippocampal neurons. Consistent with our hypothesis that AD-associated mutant (F113L) or (F131L) TOMM40 cause neurotoxicity and increase AD risk by inducing microglial activation, Taiwanese AD patients carrying *TOMM40* missense (F113L) or (F131L) variants exhibit an increased plasma level of IL-6, IL-18, IL-33, and COX-2.

In summary, the results of this study provide evidence that TOMM40 exonic variants, including rs157581 (F113L) and rs11556505 (F131L), increase the AD risk in the Taiwanese population. Further studies suggest that AD-associated mutant (F113L) or (F131L) TOMM40 cause the neurotoxicity of hippocampal neurons by inducing the activation of microglia and NLRP3 inflammasome and the release of pro-inflammatory cytokines.

## 4. Materials and Methods

### 4.1. Patients and Control Subjects

The Institutional Review Board of Chang Gung Memorial Hospital governed this study (IRB No.201700444B0C602, 201802324B0, and 202002551B0). Ethical approval for this study was granted by the IRB of Taiwan Biobank (approval number: 201506095RINC and TWBR10801-01). All participants submitted informed consent. Probable AD patients and age-matched control participants were recruited from the Department of Neurology, Chang Gung Memorial Hospital, Linkou Medical Center. AD was diagnosed according to the criteria of the recommendations from the National Institute on Aging- Alzheimer’s Association workgroups on diagnostic guidelines for Alzheimer’s disease [74]. Patients who had a Modified Hachinski ischemic score of >4 or met the NINDS-AIREN criteria for vascular dementia were excluded [75]. This study included two control groups. One of the control groups was enrolled subjects who visited CGMH for a health exam or treatment for diseases other than neurodegenerative diseases or cerebrovascular diseases. The other control group was from whole-genome sequencing (WGS) from Taiwan Biobank. Taiwan Biobank has officially developed data from community volunteers (URL: https://tai-wanview.twbiobank.org.tw/browse3, accessed on 14 February 2023).

### 4.2. DNA Extraction, WGS and Data Processing

Genomic DNA was obtained from blood samples with Gentra Puregene Blood Kit (Qiagen). After library amplification, DNA was analyzed for deep-targeted sequencing on the Ion Torrent PGM system. The panel covering *PVRL2* (GRCh38: 19:44,846,135-44,889,227), *TOMM40* (GRCh38: 19:44,891,219-44,903,688, and *APOE* (GRCh38: 19:44,905,748-44,909,394) was used to examine the genetic variants in 80 AD patients. In total, 98.5% of target bases were read more than 20 times for the depth of coverage. Sequencing data were aligned to the hg38 human reference genome and analyzed using Torrent Suite Software. Variants were filtered by Bam-Utils v1.0.2. Filtered variants were annotated using SnpEff v4.2. The in-house genetic database was used to exclude variants in 250 healthy subjects. Integrative Genome Viewer (IGV) software (http://software.broadinstitute.org/igv/, accessed on 14 February 2023) was used for mutation analysis. Allele frequency <1% in 1000 Genomes were defined as rare genetic variants. The joint variant calling file (VCF) was annotated with refGene gene regions, single-nucleotide polymorphism (SNP) effects, functional effect prediction tools, and the Exome Variant Server (EVS) and 1000 Genomes minor allele frequencies (MAFs) using Annovar (http://www.openbioinformatics.org/annovar/, accessed on 14 February 2023). Annotated VCF was analyzed as follows: Variants in exons and splice sites of *PVRL2*, *TOMM40*, and *APOE* genes were extracted with MAF of <1% in genetic databases, including the dbSNP database and the 1000 Genomes project. The variants were further interpreted and then manually annotated using the Human Gene Mutation Database (HGMD, www.hgmd.cf.ac.uk, accessed on 14 February 2023), AD&FTD (www.molgen.ua.ac.be/admutations/, accessed on 14 February 2023), AlzForum (www.alzforum.org/mutations, accessed on 14 February 2023) databases, and literature searches. Genetic variants in target genes were confirmed using Sanger sequencing.

### 4.3. Sequencing and Genotyping of TOMM40 Genetic Variants

Genetic variants and SNPs of TOMM40 identified from the cohort were further confirmed by performing TaqMan analysis or Sanger DNA sequencing (Applied Biosystems, Framingham, MA, USA). SNP rs157581, rs11556505, and rs440446 were examined using TaqMan genotyping probes (C_3084827_10/rs157581, C_2769404_10/rs11556505, C_905012_20/rs440446, Thermo Fisher Scientific, Waltham, MA, USA).

### 4.4. Cell Culture

BV2 mouse microglial cells and HT22 mouse hippocampal neuronal cells were purchased from Elabscience (Cat. EP-ML-0697 and EP-CL-0493) and maintained in a DMEM medium containing 10% FBS. Cells were grown at 37 °C in humidified air with 5% CO_2_ and then sub-cultured into different culture plates.

### 4.5. Transfection of TOMM40 Genetic Variants

The cDNA of WT, (F113L), or (F131L) TOMM40 was subcloned into the pcDNA3 expression vector (Invitrogen, Carlsbad, CA, USA) containing the FLAG (DYKDDDDK)-tag sequence. WT or mutant TOMM40 plasmids were transfected into BV2 microglial cells using the Lipofectamine 2000 transfection reagent (Thermo Fisher Scientific). After transfection for 2 or 3 days, transfected cells were used for the experiments described below.

### 4.6. Determination of Mitochondrial Membrane Potential (ΔΨm) and Mitochondrial Superoxide

To analyze ΔΨm, control or transfected BV2 microglial cells interacted with 100 nM of ΔΨm-sensitive dye tetramethylrhodamine methyl ester (TMRM; Thermo Fisher Scientific) for 30 min at 37 °C. The mitochondrial level of superoxide was measured by incubating BV2 microglial cells for 30 min at 37 °C with 5 μM MitoSox Red dye (Thermo Fisher Scientific), which is oxidized by superoxide and produces red fluorescence. TMRM or MitoSox Red images of 36 fields per well were obtained with the aid of the LionHeart FX automatic microscope (BioTek, Winooski, VT, USA). Fluorescent signals of TMRM or MitoSox Red were quantified and analyzed using Gen5 software (BioTek).

### 4.7. Immunofluorescence Staining of Iba-1

BV2 microglial cells were fixed with 4% paraformaldehyde and permeabilized with 0.5% Triton X-100. Fixed cells interacted with the anti-Iba-1 primary antibody (iReal, Hsinchu, Taiwan). Cells then interacted with the Alexa Fluor 488-conjugated secondary antibody (Invitrogen). For each imaging experiment, thirty-six images of control or transfected BV2 microglial cells were taken using the LionHeart FX automatic microscope (BioTek), and fluorescence intensity was then analyzed using Gen5 software (BioTek).

### 4.8. Immunoblotting Assays

Proteins were extracted from control or transfected BV2 microglial cells using the RIPA lysis buffer. Proteins were separated using SDS-polyacrylamide gel electrophoresis and transferred to PVDF membranes. Membranes interacted overnight at 4 °C with diluted primary antibodies (Appendix A). After washing, membranes interacted with HRP-conjugated anti-rabbit or anti-mouse secondary antibodies. Subsequently, immunoreactive proteins were detected using the ECL kit. The relative protein level was quantified by using Image J software and normalized to β-actin.

### 4.9. Measurement of Pro-Inflammatory Cytokines in Culture Medium

The level of IL-1β, IL-6, or TNF-α was measured using an ELISA kit (Abcam, Waltham, MA, USA). Briefly, 100 μL of the culture medium of control or transfected BV2 microglial cells and cytokine standards were added to 96-well pales coated with the primary antiserum. Then, the biotinylated antibody was loaded into the wells. Following 1 h incubation, the HRP-streptavidin reagent was added to the 96 wells, and OD450 was measured using a microplate reader.

### 4.10. Determination of Cell Viability of HT22 Hippocampal Neurons

The culture medium (CM) of HT22 hippocampal neuronal cells was replaced with CM of control or transfected BV2 microglial cells. Following 24-h incubation, the cell viability of HT22 hippocampal neurons was assessed using the CCK-8 assay kit (Sigma-Aldrich). Briefly, WST-8 was applied to culture wells for 1 h, and then OD at 450 nm was measured with a spectrophotometer.

### 4.11. Measurement of Plasma Levels of Cytokines or COX-2

Plasma levels of IL-1β, IL-6, IL18, IL-23, IL-33, TNF-α, and COX-2 were determined using an ELISA kit (Abcam). Briefly, 10 μL of the plasma sample was loaded into 96-well pales coated with the primary antibody at 25 °C for 150 min. Subsequently, the biotinylated secondary antiserum was added to the wells. After 1 h incubation, the HRP-streptavidin reagent was loaded into the 96 wells, and then OD450 was detected on a microplate reader.

### 4.12. Statistics

All results were expressed as the mean ± S.D. value. Demographic data for clinical subjects and the frequencies of genotypes between AD patients and control subjects were compared using an χ^2^-test (Fisher’s exact test) or *t*-test, where appropriate. All results were analyzed by using the GraphPad Prism Program and SAS software version 9.1.3. For cells’ experimental results, statistical significance was evaluated by a one-way ANOVA with Tukey’s post-hoc test (multiple groups) or unpaired two-tailed Student’s *t*-test (two groups). Statistical differences were considered significant at a *p*-value of <0.05.

## Figures and Tables

**Figure 1 ijms-24-04085-f001:**
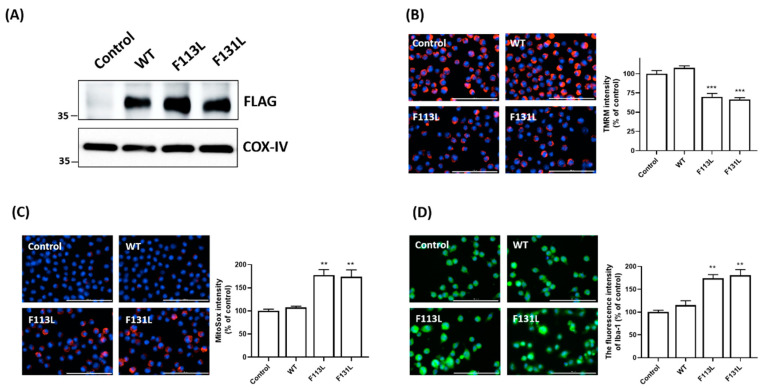
Mutant (F113L) or (F131L) TOMM40 causes dysfunction and oxidative stress of mitochondria, leading to microglial activation. (**A**) BV2 microglial cells were transfected with empty pcDNA3 vector (control), cDNA of FLAG-tagged wild-type (WT), (F113L), or (F131L) TOMM40. Immunoblotting assay showed that WT or mutant TOMM40 was expressed in the mitochondrial fraction of BV2 microglial cells. Cytochrome c oxidase subunit IV (COX-IV) was used as an internal control for mitochondrial fraction. (**B**) Compared to control cells or cells expressing WT TOMM40, decreased TMRM fluorescence intensity and depolarization of mitochondrial membrane potential were BV2 microglial cells expressing mutant (F113L) or (F131L) TOMM40. (**C**) Expression of mutant (F113L) or (F131L) TOMM40 significantly increased fluorescence intensity of MitoSox and mitochondrial superoxide level in BV2 microglial cells. (**D**) Immunofluorescence staining of microglial marker Iba-1demonstrated that expression of (F113L) or (F131L) TOMM40 led to activation of BV2 microglial cells by upregulating protein expression of Iba-1. Scale bar is 100 μm. Each bar represents mean ± S.D. value of four experiments. Each experiment was performed in triplicate. ** *p* < 0.01, *** *p* < 0.001 compared to control BV2 microglial cells.

**Figure 2 ijms-24-04085-f002:**
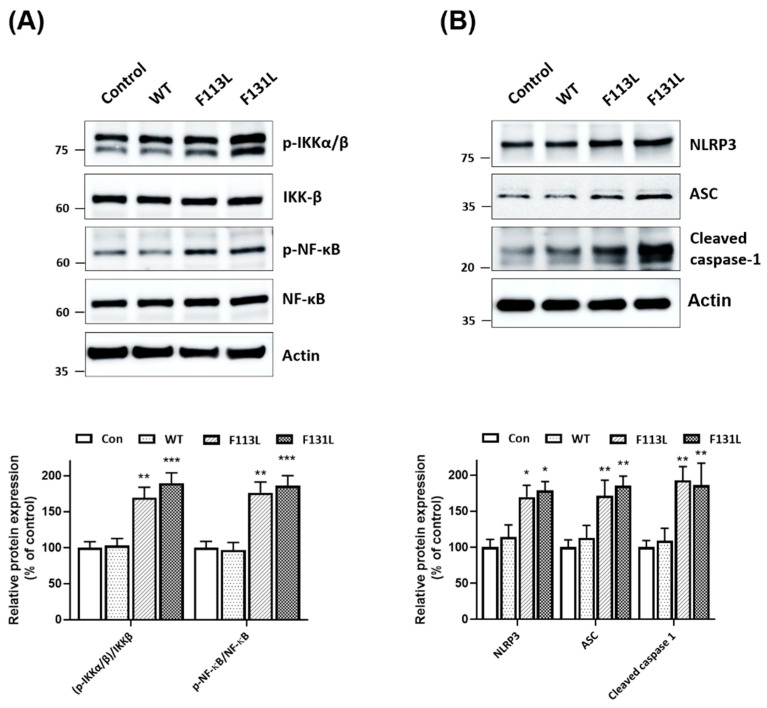
TOMM40 genetic variants induce activation of NF-κB signaling and NLRP3 inflammasome in microglial cells. (**A**) Western blot analysis showed that expression of (F113L) or (F131L) TOMM40 induced activation of NF-κB cascade by increasing protein expression of phospho-IKKα/β^Ser176/180^ and phospho-NF-κB p65^Ser536^ in BV2 microglial cells. (**B**) Compared with control or cDNA of WT TOMM40-transfected cells, transfection of cDNA encoding (F113L) or (F131L) TOMM40 caused activation of NLRP3 inflammasome by upregulating protein level of NLRP3 and ASC or cleaved caspase-1 in BV2 microglial cells. Each bar shows mean ± S.D. of four experiments. * *p* < 0.05, ** *p* < 0.01, *** *p* < 0.001 compared to control BV2 microglial cells.

**Figure 3 ijms-24-04085-f003:**
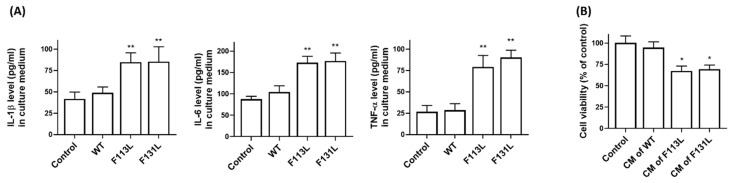
TOMM40 genetic variants induce the secretion of pro-inflammatory cytokines in microglial cells, leading to cell death of hippocampal neurons. (**A**) Compared to control cells or cells expressing WT TOMM40, BV2 microglial cells’ expression of (F113L) or (F131L) TOMM40 significantly increased secretion of pro-inflammatory IL-1β, IL-6, or TNF-α in culture medium of BV2 microglial cells. (**B**) Culture medium (CM) of HT22 hippocampal neurons was replaced with CM from BV2 microglial cells transfected with cDNA of WT, (F113L) or (F131L) TOMM40. One day after replacement, CM of BV2 microglia cells expressing mutant (F113L) or (F131L) TOMM40 significantly reduced cell viability of HT22 hippocampal neurons. Each bar represents mean ± S.D. of four experiments. Each experiment was performed in triplicate. * *p* < 0.05 or ** *p* < 0.01 compared to control BV2 microglial cells or HT22 hippocampal neurons.

**Figure 4 ijms-24-04085-f004:**
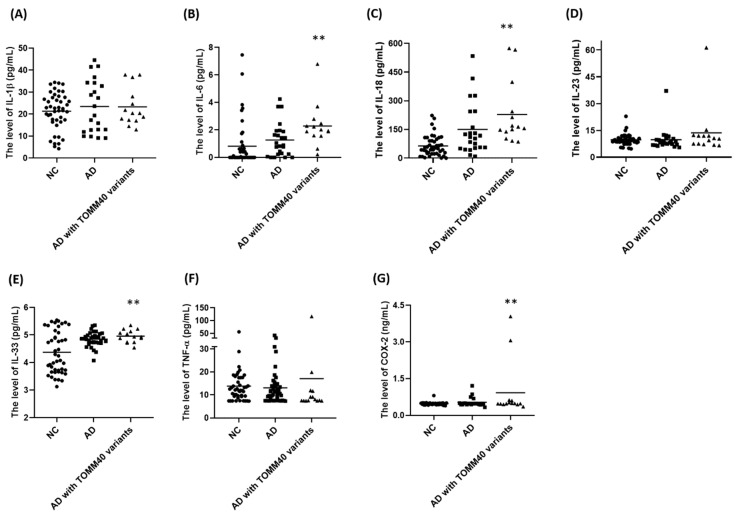
Plasma levels of IL-6, IL-18, IL-33, and COX-2 are significantly increased in AD patients carrying TOMM40 genetic variants. (**A**–**G**) ELISA assay showed that, compared to NC or AD patients, plasma levels of IL-6, IL-18, IL-33, and COX-2 were significantly upregulated in AD patients carrying TOMM40 genetic variants, including rs772262361, rs157581, and rs11556505. *** p* < 0.01 compared with NC.

**Table 1 ijms-24-04085-t001:** Genetic variants within *TOMM40-APOE* region associated with risk of AD.

Gene	SNP	Position	MAF (Cases/NC)	MAF * dbSNP	OR (95% CI)	*p* Value
*PVRL2*: Intron variant	rs394221	45368424	0.51/0.38	0.45	1.7 (1.2~2.4)	0.001
*TOMM40*:Synonymous, p.Ser66=	rs772262361	45394870	0.013/0.0	0.00004	-	-
*TOMM40*: Intron	rs184017	45394969	0.34/0.16	0.20	2.8 (2.0~4.0)	4.2 × 10^−8^
*TOMM40*: Intron	rs2075650	45395619	0.25/0.07	0.13	4.2 (2.8~6.1)	1.1 × 10^−10^
*TOMM40*: Missense, p.Phe113Leu	rs157581 *	45395714	0.38/0.23	0.23	2.1 (1.5~2.9)	4.4 × 10^−5^
*TOMM40*: Missense, p.Phe131Leu	rs11556505 *	45396144	0.26/0.10	0.11	3.3 (2.2~4.8)	2.5 × 10^−8^
*TOMM40*: Intron	rs157582	45396219	0.34/0.18	0.22	2.4 (1.7~3.3)	3.3 × 10^−6^
*APOE*: Missense, p.Asn14Lys	rs440446 *	45409167	0.56/0.33	0.38	2.6 (1.9~3.6)	1.4 × 10^−8^
*APOE*: Intron	rs769449	45412079	0.25/0.08	0.11	3.6 (2.4~5.3)	2.7 × 10^−9^

* SNPs are missense variants.

**Table 2 ijms-24-04085-t002:** Demographics of normal controls (NC), AD patients, and controls from Taiwan Biobank.

	NC	AD Patients	Controls *	*p* Value ^1^	*p* Value ^2^
Number	213	393	1025		
Age (years)	67.4 ± 10.3	74.0 ± 8.7	58.7 ± 5.4	<0.0001	<0.0001
Men, N (%)	111 (52.1)	142 (36.1)	524 (51.1)	0.0001	<0.0001
Education (years)	8.1 ± 4.5	6.2 ± 4.7	5.3 ± 1.1	<0.0001	<0.0001
Hypertension, N (%)	99 (46.5)	191 (48.6)	207 (20.2)	0.62	<0.0001
Diabetes, N (%)	153 (71.8)	271 (69.0)	81 (7.9)	0.46	<0.0001
Global cognition, MMSE	24.3 ± 5.4	16.5 ± 6.5	27.1 ± 2.6	<0.0001	<0.0001
*APOE* ε4 carrier, N (%)	12 (5.63)	201 (51.2)	166 (16.2)	<0.0001	<0.0001
*TOMM40*					
rs772262361, p.Ser66 = AA/AG (%)	100/0	99.5/0.5	-	0.24 **	-
rs157581, p.Phe113Leu TT/TC/CC (%)	67.3/30.3/2.4	40/49.9/10.1	59.9/35/5.1	<0.0001	<0.0001
rs11556505, p.Phe131Leu CC/CT/TT (%)	92.6/7.4/0	54.4/40/5.6	81.9/17.3/0.9	<0.0001	<0.0001

Data are expressed as percentages or mean ± S.D. * Controls from Taiwan Biobank were from the whole-genome sequencing (WGS) database from Taiwan Biobank. Comparisons were analyzed using χ^2^-tests (** Fisher’s exact test) or *t*-tests where appropriate. *p*-value ^1^: NC versus AD; *p*-value ^2^: Control from Taiwan Biobank versus AD.

**Table 3 ijms-24-04085-t003:** The plasma level of cytokines or COX-2 between NC and AD.

	NC (n = 45)	AD (n = 37)	AD with *TOMM40* Genetic Variants (n = 14)	*p* Value
Age (years)	72.1 ± 9.0	72.9 ± 6.0	71.29 ± 6.3	0.6740
Men/Female	21/24	14/23	4/10	0.4275
APOE ε4 carrier (%)	0	32.4	85.7	<0.0001
IL-1β (pg/mL)	21.4 ± 8.4	23.1 ± 10.8	23.3 ± 8.6	0.3884
IL-6 (pg/mL)	0.8 ± 1.6	1.6 ± 1.5	2.3 ± 1.6	0.0237
IL-18 (pg/mL)	64.4 ± 53.0	193.5 ± 200.6	228.66 ± 164.7	<0.0001
IL-23 (pg/mL)	9.7 ± 3.2	10.5 ± 8.9	13.7 ± 13.9	0.5867
IL-33 (pg/mL)	4.4 ± 0.8	4.9 ± 0.3	5.0 ± 0.2	0.0005
TNF-α (pg/mL)	13.8 ± 8.1	14.5 ± 18.9	17.1 ± 28.8	0.9342
COX-2 (ng/mL)	0.5 ± 0.06	0.7 ± 0.7	0.9 ± 1.1	0.0453

Data are expressed as percentages or mean ± S.D.

## Data Availability

All data generated or analyzed during this study are included in this published article.

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
