# Peer review of "TOMM40 Genetic Variants Cause Neuroinflammation in Alzheimer’s Disease"

_ijms, 2023, doi:10.3390/ijms24044085_

Round 1

Reviewer 1 Report

The authors did great research on neuroinflammation due to TOMM40. I have some suggestions to improve the manuscript.

The abstract is well written.

The introduction is with concrete background and clear hypothesis. 

Results are not clear in all of the figures, please improve all of the figures by clearly mentioning the staining beside the figures, enlarging the text, putting a bar scale with images so that the readers can easily get an idea about the cell size, figure-3 is very small and text is not visible.

The discussion is very small, and needs to correlate with the data obtained. There is no conclusion.

Please improve the materials and method section by providing more information so that it could be reproduced by anyone in any lab. For example, 4.2 data were analyzed by different software, which does not give any idea how these were done, what was the criteria, what was the technique, or whether these were done correctly or not. Please provide more clear descriptions for all of them.

Please include a conclusion based on the data obtained.

Please include a graphical abstract or research summary based on the data.

Author Response

The followings are the reply to comments given by Reviewer #1 and the revisions I have made based on Referee #1’s comments (labeled in red in the revised manuscript):

(1) As requested by Reviewer #1, we have improved the quality of all figures by enlarging the text and size of Figures and adding bar scales within images (Please see Figures and Figure Legends of revised manuscript.).

(2) As instructed by Referee #1, we have rewritten the Discussion section by correlating with the data obtained (Please see Discussion section of revised manuscript.).

(3) As requested by Reviewer #1, conclusion of this study has been added in the end of Discussion section (Please see Discussion section of revised manuscript.).

(4) As instructed by Referee #1, we have improved the Materials and Methods section by providing more detailed information (Please see Materials and Methods section of revised manuscript.).

(5) As requested by Reviewer #1, graphical abstract of this study has been added in the revised manuscript.

Reviewer 2 Report

This manuscript reports novel results indicating that TOMM40 variants cause inflammatory features in microglial and neuronal cell lines and in AD patients. The authors use multiple approaches (genetic, molecular, immunohistochemistry) to obtain data supporting this conclusion. Although the experimental work appears to be well designed and done, important information is missing from the manuscript. Therefore, the following comments should be addressed in order to improve it.

1.    A paragraph or two should be included in Material and Methods describing how image analysis was performed. How many images were analyzed per biological replicate?

2.    It is not clear what the authors consider to be the “n” value: the number of experiments? the number of biological replicates?  Please, clarify this important point.

3.    The Discussion is surprisingly superficial and it should be improved.

Author Response

The followings are the reply to comments given by Referee #2 and the revisions I have made based on Reviewer #2’s comments (labeled in red in the revised manuscript):

(1) As requested by Referee #2, detailed procedures of imaging analysis has been described in Materials and Methods section (Please see Materials and Methods section of revised manuscript.).

(2) Reviewer #2 asked us to clarify the “n” value of experiments conducted in this study. As mentioned in Figure Legends of Figure 1 and Figure 3, each bar represents mean ± S.D. value of four experiments. Each experiment was performed in triplicate.

(3) As instructed by Referee #2, we have rewritten the Discussion section by correlating with the data obtained and adding conclusion of this study (Please see Discussion section of revised manuscript.).

Round 2

Reviewer 1 Report

The authors made the appropriate changes and now the manuscript is improved. I would recommend to publish this version.

Reviewer 2 Report

The manuscript is improved after revision. The authors have addressed my requests and comments.